# Innovative C-Arm-Free Navigation Technique for Posterior Spinal Fixation for Atlantoaxial Subluxation: A Technical Note

**DOI:** 10.3390/medicina59010011

**Published:** 2022-12-20

**Authors:** Masato Tanaka, Naveen Sake, Dae-Geun Kim, Shinya Arataki, Dhvanit Desai, Yoshihiro Fujiwara, Taro Yamauchi

**Affiliations:** 1Department of Orthopaedic Surgery, Okayama Rosai Hospital, Okayama 702-8055, Japan; 2Department of Orthopaedics, Pondicherry Institute of Medical Sciences, Kalathumettupathai, Pondicherry 605014, India

**Keywords:** atlantoaxial subluxation, Goel technique, C1 lateral mass screw, navigation

## Abstract

*Study design*: Technical note. *Objectives*: To present a novel C-arm-free technique guided by navigation to insert and place a C1 lateral mass screw. *Background and Objectives*: Atlantoaxial subluxation (AAS) is a relatively common sequelae in patients with rheumatoid arthritis (RA) and upper cervical trauma. If they present with severe symptoms, surgical intervention such as posterior fusion is indicated. The established treatment for AAS is fixation with a C1 lateral mass screw and C2 pedicle screw (modified Goel technique) to achieve bony fusion. However, this technique requires fluoroscopy for C1 screw insertion. To avoid exposing the operating team to radiation, we present here a novel C-arm-free C1 lateral mass screw insertion technique for AAS. *Materials and Methods*: A 67-year-old man was referred to our hospital with neck pain, quadriparesis, and clumsiness and numbness of both upper and lower limbs. He had undergone C3–6 posterior fusion previously in another hospital. In physical examination, he had severe muscle weakness of bilateral upper limbs and hypoesthesia of all four limbs. He had hyper-reflexia of bilateral lower limbs and pollakiuria. His Japanese orthopedic score was 8 points out of 17. Preoperative radiograms showed AAS with an atlantodental interval (ADI) of 7 mm. MRI indicated retro-odontoid pseudotumor and severe spinal cord compression at the C1–2 level. The patient underwent posterior atlantoaxial fixation under navigation guidance. To prevent epidural bleeding during the insertion and placement of a C1 lateral mass screw, we have here defined a novel screw insertion technique. *Results*: The surgical time was clocked as 127 min and blood loss was 100 mL. There were no complications per-operatively or in the postoperative period. The patient showed almost full recovery (JOA 16/17) at two months follow-up and a solid bony fusion was noticed in the radiograms at one year follow-up. *Conclusions*: This novel surgical procedure and C1 lateral mas screw placement technique is a practical and safe method in recent advances of AAS treatment. Procedurally, the technique helps prevent epidural bleeding from the screw entry point and also allows for proper C1 screw insertion under navigation guidance without exposing surgeons and staff to the risk of fluoroscopic radiation.

## 1. Introduction

Atlantoaxial subluxation (AAS) is a relatively common sequelae in rheumatoid arthritis (RA) and in patients who sustained upper cervical trauma. For the patients presenting with worsening symptoms, surgeries such as posterior fusion are indicated. For AAS, the goal of fixation is to achieve a solid bony union [1]. There are several techniques to perform posterior atlantoaxial fusion (PAAF) [2,3,4,5]. Biomechanically, the C1/2 transarticular (Magerl) screw is the strongest anchor for this purpose [6], but this procedure requires preoperative altantoaxial reduction and has a high probability of vertebral artery injury [7]. Ideally, a C2 lamina screw and a C1 lateral mass screw (LMS) fixation is considered very safe but the associated incidence of pseudoarthrosis is higher than other methods [8].

The modified Goel technique, which proposed C1 lateral mass screws combined with C2 pedicle screws, is an established surgical option [9,10,11]. Navigation-mapped spine surgery is now a widely accepted and available technology-aided procedure for placing cervical pedicle screws [12,13]. However, it usually requires fluoroscopy to place the C1 LMS precisely. To avoid exposing the operating team to radiation, we present here an innovative C-arm-free, O-arm-navigated surgical procedure and novel C1 lateral mass screw insertion technique for AAS.

## 2. Case Presentation

This study was approved by the ethics committee of our institute (27 September 2022, No. 351). The necessary consents were obtained from the patient.

### 2.1. Patient History

A 67-year-old man was referred to our hospital with complaints of neck pain, quadriparesis. He had undergone C3–6 posterior fusion in another hospital 7 years ago due to gait disturbance. He had experienced almost full recovery for 5 years. Gradually, he had clumsiness and numbness of both upper and lower limbs these past two years.

### 2.2. Physical Examination

In the examination, he presented with severe muscle weakness of bilateral upper limbs and hypoesthesia of both upper and lower limbs. His manual muscle test (MMT) scores were grade 2 for bilateral deltoid muscle and grade 4 for biceps and triceps muscle, and hyper-reflexia was detected in both upper and lower limbs. He was unable to use chopsticks, had pollakiuria, and clinically presented with a Japanese orthopedic association (JOA) score that was 8 points out of 17.

### 2.3. Preoperative Imaging

Preoperative radiograms showed AAS with an atlantodental interval (ADI) of 7 mm (Figure 1A,B). Preoperative magnetic resonance imaging (MRI) indicated severe spinal cord compression and retro-odontoid pseudotumor at C1–2 (Figure 1C,D). Sagittal reconstruction computed tomography (CT) revealed C5–6 was fused and showed narrowing of multiple disc spaces (Figure 2A). Cervical vascular imaging showed neither any abnormality of vertebral artery (VA) nor high riding VA (Figure 2B–D).

### 2.4. Surgery

The patient underwent posterior atlantoaxial fixation under navigation guidance. To prevent epidural bleeding, to insert a C1 lateral mass screw, we made use of a novel screw insertion technique. We did not choose C1–6 fusion because we wanted to preserve the neck motion of the patient as much as possible. The surgical time was clocked as 127 min and blood loss was 100 mL. The patient experienced no postoperative complications such as CSF leakage, surgical site infection, or neurological compromise.

### 2.5. Postoperative Imaging

Postoperative radiograms showed good upper cervical alignment (Figure 3A,B) Postoperative CT showed adequate screw placement and purchase at both C1 LMSs and C2 pedicle screws (Figure 3C–H).

### 2.6. One Year Follow-Up

The patient recovered almost fully (JOA 16/17) at two months follow-up. His MMT scores were grade 5 for bilateral deltoid muscle and grade 5 for biceps and triceps muscle. He became able to use chopsticks and had no pollakiuria. His hyper-reflexia and hypoesthesia in both upper and lower limbs were reduced. The retro-odontoid tumor showed a decrease in morphology and spinal cord compression was minimized at four months follow-up (Figure 4A,B).The solid bony fusion was obtained at one year follow-up (Figure 4C,D).

## 3. Operative Procedure

### 3.1. Patient Positioning and Exposure

The patient was positioned prone with the neck flexed position on a Jackson frame with a full carbon skull clamp to enable the O-arm scan. The procedure was performed under neuromonitoring. The atlas and axis were exposed with a 5 cm posterior midline incision. The reference frame for the navigation was fixed at the C2 spinous process. The O-arm was then positioned, and three-dimensional (3-D) reconstructed images were obtained, transmitted to the StealthStation navigation system Spine 7^R^ (Medtronic Sofamor Danek, Minneapolis, MN, USA), and integrated.

### 3.2. C2 Pedicle Screw

After verifying every navigation-mapped spinal instrument, the accuracy of the same should be checked and confirmed. The entry point for the C2 pedicle screw was marked and a high-speed burr was used to make the cortical breach without pushing the C2 or further causing an anterior compression the spinal cord. A navigated pedicle probe was used to ascertain the trajectory and appropriate tapping was conducted, following which the C2 pedicle screw was inserted (Figure 5).

### 3.3. C1 LMS

The entry point for the C1 lateral mass screw (LMS) was the mid-point (8 mm anterior from posterior arch and caudal aspect) of the C1 posterior arch (Figure 6A,B). A Penfield’s retractor was used to separate the inferior aspect of the posterior arch and posterior atlantoaxial venous plexus, and to retract the venous plexus inferiorly. Then, a navigated high-speed burr with a 2 mm tip was utilized to make an entry point with meticulous care of the venous plexus (Figure 6C,D). Per-operatively, caudal retraction for safeguarding the epidural veins may be taxing. After making a small hole, a navigated pedicle probe was used to enlarge the hole and penetrate the anterior aspect of the cortex of the C1 anterior arch (Figure 7A,B). The navigated tap was then carefully used, avoiding injury to the venous plexus. Finally, a C1 LMS was inserted with a bicortical purchase (Figure 7C,D). Intraoperatively, another O-arm image was obtained to confirm the position and length of the screws.

### 3.4. Reduction and Bone Graft

First, AAS was reduced by changing the neck position from flexion to extension by rotating the Mayfield skull clamp. Second, appropriately sized rods were applied over both sides and the two screws were compressed carefully. When adequate reduction was obtained, the set screws were tightened. An autologous iliac crest bone graft was placed interlminae for osteoinduction. Postoperative radiograms were taken to confirm the correct reduction and placement. A Philadelphia collar was applied to immobilize the neck.

## 4. Discussion

AAS is defined as the malalignment with increased mobility between the atlas and the odontoid process, which indicates laxity/rupture of the transverse ligament and loss of its stability. AAS is a common sequela of rheumatoid arthritis (RA) and has an incidence of 10% to 55% in RA patients [14]. It is also known to be associated with the chronic inflammatory conditions of the nasopharynx such as adenotonsilitis, retropharyngeal abscess [15], eosinophilic granuloma [16], and certain syndromic conditions such as Marfan’s syndrome [17], Down’s syndrome [18], and Klippel Feil’s syndrome [19]. In adults, it is most commonly seen in trauma to the upper cervical spine [20] and in some cases of ankylosing spondylitis [21]. Clinical presentation ranges from asymptomatic to progressively severe symptoms depending on the pathophysiology. Occipital headache and neck pain are frequently seen [22]. AAS occasionally causes intermittent and progressive cervical spinal cord compression, which presents clinically as sensory disturbances, motor weakness, and even paralysis of the limbs [23].

Diagnosis is usually made with plain lateral cervical radiograms; however, dynamic views are required to confirm instability. ADI is one of the main diagnostic factors for AAS, which is determined by measuring the distance between the posterior edge of the anterior arch of C1 and the anterior edge of dens. ADI more than 3 mm in children and more than 5 mm in adults are regarded as abnormal [24]. If a subluxation is suspected, or if the patient is presenting with symptoms of cord compromise, MRI should be conducted. CT reconstruction is also recommended. Conservative management aims at relieving the symptoms and influencing the natural course of AAS, as pain is the most common symptom of the patients suffering from AAS. Medication, cervical collar, and monitored exercises have shown good results with a decrease in complaints of pain [25]. Ranawat grading of pain and neurological status is a common criterion to consider in diagnosis and the progressive neurological changes [26].

Surgical indications of the AAS include patients presenting with symptoms of progressive cord compression, chronic myelopathy, and neuralgia not relived by conservative management. The Japanese orthopaedic association (JOA) score is used to evaluate the symptoms of cervical myelopathy in patients with AAS, which is highly reliable [27]. In our opinion, clinically severe symptoms such as JOA score < 10 points or radiological parameters such as ADI > 10 mm or space available for cord (SAC) < 14 mm indicate a surgical intervention. AAS was treated by surgical C1–C2 fusion using posterior wiring and was first described by Gallie [28]. The introduction of a transarticular screw by Magerl, providing a stable upper cervical fusion, was well received, though the procedure required preoperative reduction of the subluxation [2]. Surgical fixation with plate and screw by Goel et al. [3] was modified and is widely known as the Harms–Goel technique. The Goel–Harms technique is the fixation of C1 lateral mass screws (LMSs) and polyaxial C2 pedicle screws with bilateral rods [5]. With the modified Goel technique, the subluxation is reduced by instruments and neck extension before fixation. The fixation of the AAS has now been in the spotlight for innovative ways to approach surgically, not only for the treatment but also for better outcomes with lesser complications [29].

According to Guo et al. [30], the mean operative time was 153.9 ± 73.9 min, and the mean blood loss was 219.1 ± 195.6 mL for C1–C2 posterior fixation. In our case, the surgical time was 127 min and volume of bleeding was about 100 mL. With our novel technique, we have established that it is not necessary to expose the C1 lateral mass directly like Tan’s technique and that the blood loss and the surgical time can be reduced. The C-arm fluoroscopy navigation system can reduce the amount of the radiation exposure per unit time compared with O-arm navigation [31]. However, when the quality of the images and total radiation dose are considered, O-arm navigation is more useful.

With the current advancements in technology, complex cervical spine surgeries are nowadays easily approachable [32]. We have introduced a novel technique of placing C1 LMS using an O-arm-navigation-guided technique without utilizing a C-arm (Figure 8). The advantage of our mid-point technique is that there is less bleeding and easy exposure to the available screw entry points in the cases that present with a requirement for surgical fixation. With the Goel–Harms or Notch techniques, the atlantoaxial venous plexus should be exposed and retract caudally, and this step sometimes causes massive bleeding. However, there are many cases, which have a contraindication of a C1 posterior arch screw (Tan’s technique) because of the thin posterior arch of the C1 [11]. If the screw penetrates the cranial side of the posterior arch, it is a high risk of VA injury. The over-the-arch technique reported by Lee et al. is also a little dangerous to injure the VA because the entry point of the screw is very near to the VA [33]. With our new technique, a clear operation field is achieved by a retraction of the posterior half of the epidural venous plexus. Three-dimensional reconstructed imaging by the O-arm-navigated systems provides accurate and precise angles, along with screw trajectories for the introduction and placements of the C1 LMS (Figure 6). The entry point and the trajectory as described is the safest pathway for the insertion and advancement of the screw, giving the best possible stability to the fixation (Figure 6 and Figure 7). Navigation also provides a precise length of the LMS, with a proper bicortical purchase and an avoidance of vascular injuries. The usage of a navigated high-speed burr allows for a decrease in the compressive downforce on the cord throughout the LMS placement. In this novel technique, efforts have been made to present a procedure that is radiation-free to the operating team, from the image intensifier, and to utilize the availability of O-arm-navigation-mapped spinal guidance systems to achieve a precise, biomechanically stable C1–2 bony fusion without any complications.

Though there are numerous benefits of the above technique, it does have a few limitations. The availability of the O-arm navigation system may be sparse by region. Navigated screw placement requires training of the operating personnel. The placement of the reference frame may require care throughout the surgery, displacement of which would require a rescan with the O-arm or cause error in the screw insertion.

## 5. Conclusions

This novel procedure is a useful and safe technique. It allows for ideal points for the insertion and placement of screws without the risk of VA injury and venous plexus under navigation guidance. This technique is performed without exposing surgeons and staff to the risk of radiation.

## Figures and Tables

**Figure 1 medicina-59-00011-f001:**
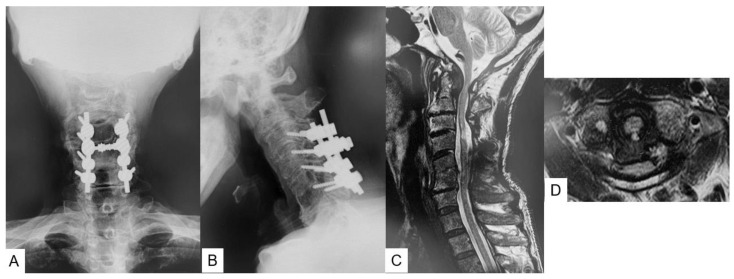
Preoperative images, (**A**): antero-posterior radiogram, (**B**): lateral flexion radiogram, ADI = 7 mm, (**C**): mid-sagittal T2-weighted MRI, (**D**): axial T2-weighted MRI at C1–2.

**Figure 2 medicina-59-00011-f002:**
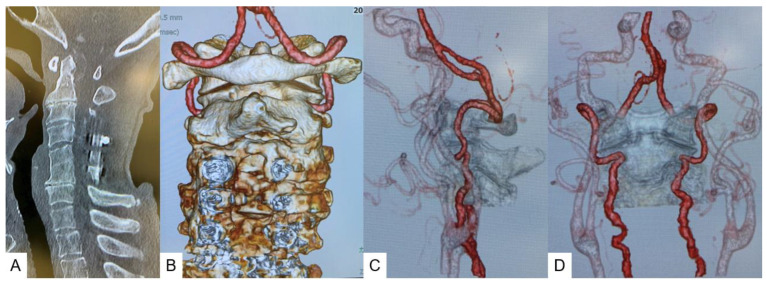
Preoperative CT and vascular images, (**A**): mid-sagittal reconstruction CT, (**B**): posteroanterior 3D CT image including arteries, (**C**): lateral VA image, (**D**): posteroanterior VA image.

**Figure 3 medicina-59-00011-f003:**
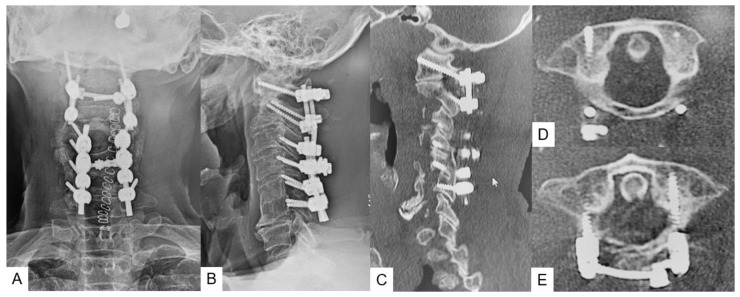
Postoperative images, (**A**): anteroposterior radiogram, (**B**): lateral radiogram, (**C**–**E**) CT, C: C1 LMS (mid-point), (**D**,**E**): C1 axial view.

**Figure 4 medicina-59-00011-f004:**
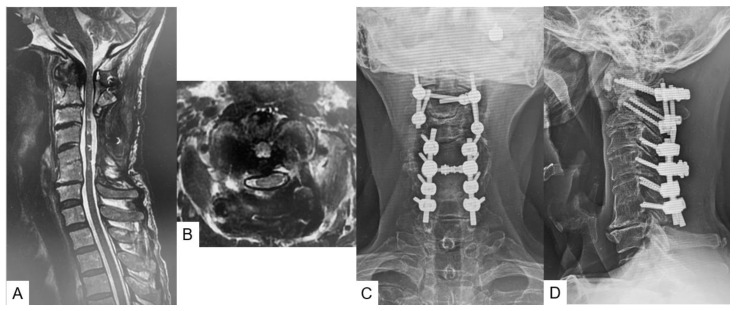
Follow-up images, (**A**): mid-sagittal T2-weighted MR imaging, (**B**): axial T2-weighted MR imaging at C1–2, (**C**): anteroposterior radiogram, (**D**): lateral radiogram.

**Figure 5 medicina-59-00011-f005:**
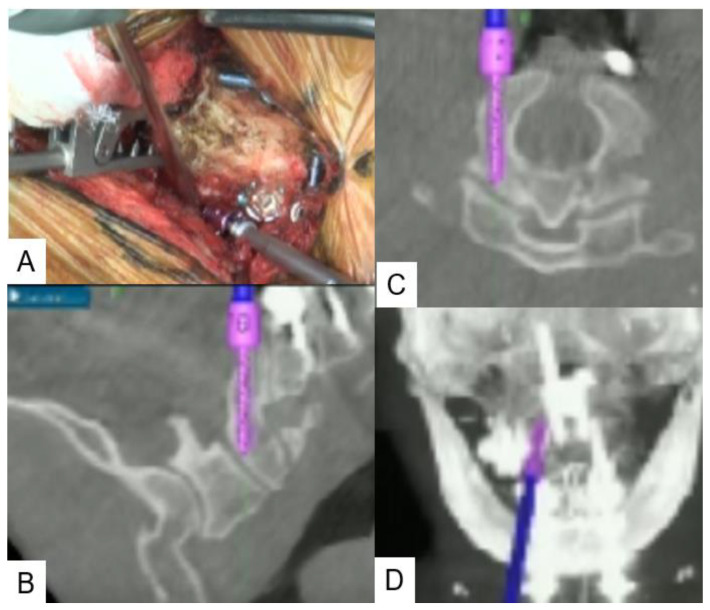
C2 pedicle screw (**A**): intra-operative image, (**B**): sagittal view, (**C**): axial view, (**D**): coronal view.

**Figure 6 medicina-59-00011-f006:**
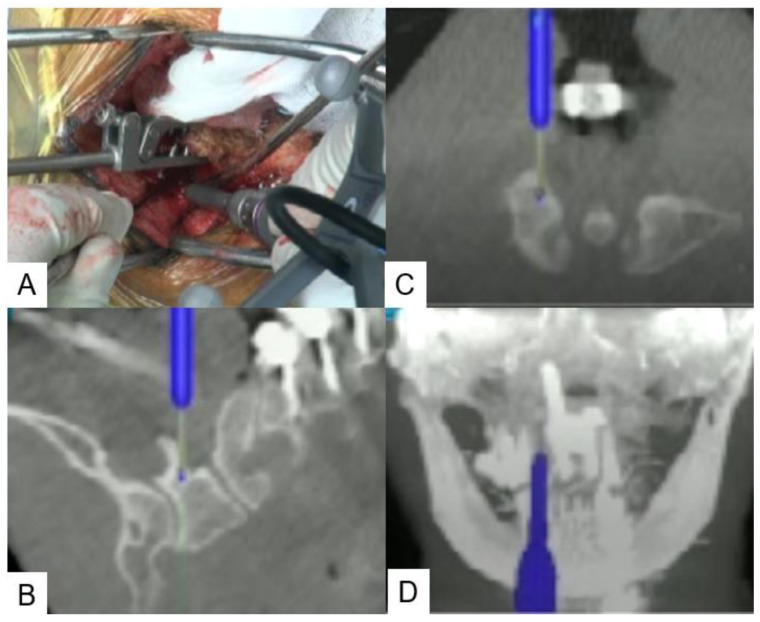
C1 lateral mass screw (**A**): intraoperative image, (**B**): sagittal view, (**C**): axial view, (**D**): coronal view.

**Figure 7 medicina-59-00011-f007:**
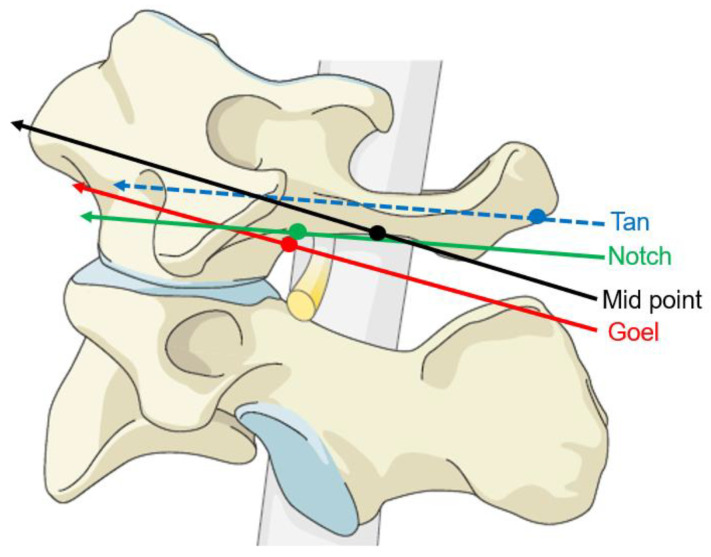
Lateral view of entry point and trajectory of four kinds of LMS placement. Blue line: Tan method, green line: Notch method, red line: Goel method, black line: mid-point method. Every circle indicates bony entry point.

**Figure 8 medicina-59-00011-f008:**
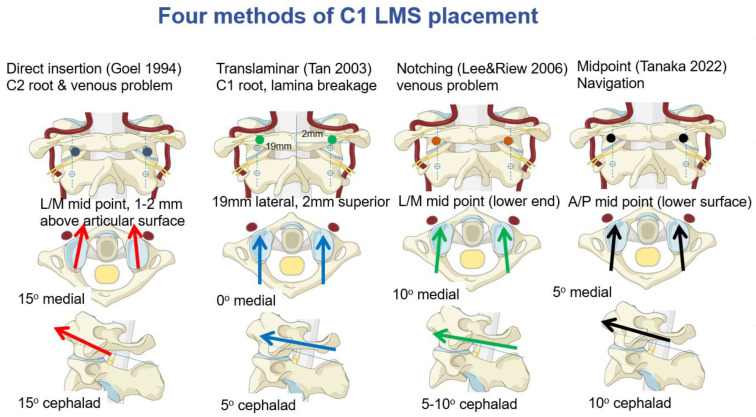
Four methods of C1 LMS placement [3,7,11]. red arrows, screw direction of Goel method; blue arrows, screw direction of Tan method; green arrows, screw direction of Noch method; black arrows, screw direction of Midpoint (our) method.

## Data Availability

The data presented in this study are available in the article.

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
