# Peer review of "Innovative C-Arm-Free Navigation Technique for Posterior Spinal Fixation for Atlantoaxial Subluxation: A Technical Note"

_medicina, 2022, doi:10.3390/medicina59010011_

Round 1

Reviewer 1 Report

The innovative technique, for posterior fusion of atloaxial subluxation, is well explained and well documented with radiographic images as well as very clear anatomical drawings.

Unfortunately, I have to say that: in my experience the work seems written, with great interest and technical competence, but without clinical interest.

To be clearer: I found a hasty and incomplete draft on page 2 lines 59,60,61, where, in my opinion, it was necessary to specify how long the patient had been quadriparetic and how long ago he had been operated on with instrumented posterior fusion C3C6.

I noticed another major defect on page 6, lines 143-148, where, after the above excellent description of the screw insertion technique. The reduction of the subluxation is described, by anchoring two bars to the C1C2 screws, plus the arthrode by iliac crest bone graft (without explaining whether autologous or bank homologous).

At this point a C2C3 discontinuity is created.

The paper does not explain what has been done to avoid a C2C3 junctional instability.

In the post-op Rx images I do not see any linking "dominoes" between bars C1C2 and C3C6. Was an arthrodesis with bone graft extended after bleeding the C2C3 joints bilaterally?

In my clinical and surgical experience, these aspects are fundamental.

Author Response

Dear respective reviewer,

We appreciate your important contribution.

Unfortunately, I have to say that: in my experience the work seems written, with great interest and technical competence, but without clinical interest.

Thank you for your comment. We have experienced relative difficulty to perform atlantoaxial posterior fusion because of bleeding from posterior C1/2 venous plexus.

As you know, there are several C1 lateral mass screw fixation techniques. Each technique has its own advantages and disadvantages. We think that notch technique is one of the most popular techniques to put C1 lateral mass screw. This is the modification of that technique with navigation to reduce blood loss and irritation of C2 nerve root.

To be clearer: I found a hasty and incomplete draft on page 2 lines 59,60,61, where, in my opinion, it was necessary to specify how long the patient had been quadriparetic and how long ago he had been operated on with instrumented posterior fusion C3C6.

We appreciate your important comment. We changed the sentences as follows.

A 67-year-old man was referred to our hospital with complaints of neck pain, quadriparesis. He had undergone C3-6 posterior fusion in another hospital 7 years ago due to gait disturbance. He had got almost full recovery for 5 years. Gradually, he had clumsiness and numbness of both upper and lower limbs these two years.

The reduction of the subluxation is described, by anchoring two bars to the C1C2 screws, plus the arthrode by iliac crest bone graft (without explaining whether autologous or bank homologous).

Thank you for your comment. We changed the sentence as follows.

An autologous iliac crest bone graft was placed interlminae for osteoinduction.

At this point a C2C3 discontinuity is created.

The paper does not explain what has been done to avoid a C2C3 junctional instability. In the post-op Rx images I do not see any linking "dominoes" between bars C1C2 and C3C6. Was an arthrodesis with bone graft extended after bleeding the C2C3 joints bilaterally?

Thank you for your valuable question. We’d like to preserve the neck motion of this patient as much as possible. And this was also discussed with the patient beforhand. As you mentioned, C1-C6 fusion was also a good option for this patient. We added the sentence as follows.

We didn’t choose C1-6 fusion because we’d like to preserve the neck motion of the patient as much as possible.

Reviewer 2 Report

·        Line 67 – Why do you describe increased urinary frequency in this patient as Pollakiuria?

·        Line 96 – 2.6 one-year follow-up - please provide follow-up outcomes as in section 2.2: including MMT / reflexes / sensation / use of chopsticks and pollakiuria

·        Line 191 – I may have missed the point here - You describe a reduced surgical time [127 mins compared to 153 mins] plus reduced blood loss. I think a discussion is required relating to increased exposure using O arm (O-arm navigation system use is shorter in radiation time and larger in radiation exposure than C-arm fluoroscopy navigation system. However, the amount of the radiation exposure per unit time in O-arm navigation system is larger than in C-arm fluoroscopy navigation system (Maruo et al, 2017- https://journals.sagepub.com/doi/10.1055/s-0036-1582933).

Author Response

Dear respective reviewer,

We appreciate your important contribution.

Line 67 – Why do you describe increased urinary frequency in this patient as Pollakiuria?

Thank you for your comment. Elder male patients sometimes had a benign prostate hypertrophy and pollakiuria. As you mentioned, pollakiuria isn’t always related to myelopathy symptom. However, this patient complained pollakiuria, so we just described in the physical examination part.

Line 96 – 2.6 one-year follow-up - please provide follow-up outcomes as in section 2.2: including MMT / reflexes / sensation / use of chopsticks and pollakiuria.

His MMT scores were grade 5- for bilateral deltoid muscle, grade 5 for biceps and triceps muscle. He became able to use chopsticks and no pollakiuria. His hyper-reflexia and hypoesthesia in both upper and lower limbs were reduced.

Line 191 – I may have missed the point here - You describe a reduced surgical time [127 mins compared to 153 mins] plus reduced blood loss. I think a discussion is required relating to increased exposure using O arm (O-arm navigation system use is shorter in radiation time and larger in radiation exposure than C-arm fluoroscopy navigation system. However, the amount of the radiation exposure per unit time in O-arm navigation system is larger than in C-arm fluoroscopy navigation system (Maruo et al, 2017- https://journals.sagepub.com/doi/10.1055/s-0036-1582933).

We appreciate your valuable comment. According to your advice, we added the sentences as follows in discussion part.

C-arm fluoroscopy navigation system can reduce the amount of the radiation exposure per unit time compared with O-arm navigation [X]. However, the quality of the images and total radiation dose are considered, O-arm navigation is more useful.

Round 2

Reviewer 1 Report

The revised version is ok